# An outbreak of acute jaundice syndrome (AJS) among the Rohingya refugees in Cox's Bazar, Bangladesh: Findings from enhanced epidemiological surveillance

Md Khadimul Anam Mazhar [1]*, Flavio Finger [2,3,4], Egmond Samir Evers[1],
Anna Kuehne[2,5,6,7], Melissa Ivey[8], Francis Yesurajan[1], Tahmina Shirin[9], Nurul Ajim[8],
Ahammadul Kabir[1], Jennie Musto[1], Kate White[8], Amrish Baidjoe[1,2,6,10], Olivier le Polain
de Waroux[2,5,6,7]

**1** World Health Organization, Cox's Bazar Emergency Sub-Office, Cox's Bazar, Bangladesh, **2** Global
Outbreak Alert and Response Network, Geneva, Switzerland, **3** Centre for the Mathematical Modelling of
Infectious Diseases, London School of Hygiene and Tropical Medicine, London, United Kingdom,
**4** Department of Infectious Disease Epidemiology, London School of Hygiene and Tropical Medicine,
London, United Kingdom, **5** Public Health England, Letchworth Garden, United Kingdom, **6** London School of
Hygiene and Tropical Medicine, London, United Kingdom, **7** UK-Public Health Rapid Support Team, United
Kingdom, **8** Médecins sans Frontières, Bangladesh, **9** Institute of Epidemiology, Disease Control and
Research (IEDCR), Dhaka, Bangladesh, **10** International Committee of the Red Cross, Geneva, Switzerland

* khadimulmazhar@gmail.com

doi.org/10.1371/journal.pone.0250505

Brody School of Medicine, UNITED STATES

**Data Availability Statement:** All data used in this
study are owned by a third-party stakeholder,
Ministry of Health & Family Welfare (MoHFW). A

## Abstract

In the summer of 2017, an estimated 745,000 Rohingya fled to Bangladesh in what has
been described as one of the largest and fastest growing refugee crises in the world. Among
numerous health concerns, an outbreak of acute jaundice syndrome (AJS) was detected by
the disease surveillance system in early 2018 among the refugee population. This paper
describes the investigation into the increase in AJS cases, the process and results of the
investigation, which were strongly suggestive of a large outbreak due to hepatitis A virus
(HAV). An enhanced serological investigation was conducted between 28 February to 26
March 2018 to determine the etiologies and risk factors associated with the outbreak. A total
of 275 samples were collected from 18 health facilities reporting AJS cases. Blood samples
were collected from all patients fulfilling the study specific case definition and inclusion crite-
ria, and tested for antibody responses using enzyme-linked immunosorbent assay (ELISA).
Out of the 275 samples, 206 were positive for one of the agents tested. The laboratory
results confirmed multiple etiologies including 154 (56%) samples tested positive for hepati-
tis A, 1 (0.4%) positive for hepatitis E, 36 (13%) positive for hepatitis B, 25 (9%) positive for
hepatitis C, and 14 (5%) positive for leptospirosis. Among all specimens tested 24 (9%)
showed evidence of co-infections with multiple etiologies. Hepatitis A and E are commonly
found in refugee camps and have similar clinical presentations. In the absence of robust
testing capacity when the epidemic was identified through syndromic reporting, a particular
concern was that of a hepatitis E outbreak, for which immunity tends to be limited, and which
may be particularly severe among pregnant women. This report highlights the challenges of
identifying causative agents in such settings and the resources required to do so. Results

written approval was obtained from the MoHFW to use this anonymised data set only for the purpose of sharing the results of this outbreak investigation and in the context of this manuscript. Any further request to access the data has to be approved by the MoHFW responsible for overseeing the Rohingya refugee response. The contact details are given below: - Civil Surgeon Office, Ministry of Health & Family Welfare, - Kolatoli Road, Cox's Bazar, Bangladesh - Email: coxsbazar@cs.dghs.gov.bd.

**Funding:** The authors received no specific funding for this work. The UK Public Health Rapid Support Team is funded by UK Aid from the Department of Health and Social Care and is jointly run by Public Health England and the London School of Hygiene & Tropical Medicine. The views expressed in this publication are those of the authors and not necessarily those of the National Health Service, the National Institute for Health Research or the Department of Health and Social Care. The funders had no role in study design, data collection and analysis, decision to publish, or preparation of the manuscript.

**Competing interests:** The authors have declared that no competing interests exist.

from the month-long enhanced investigation did not point out widespread hepatitis E virus (HEV) transmission, but instead strongly suggested a large-scale hepatitis A outbreak of milder consequences, and highlighted a number of other concomitant causes of AJS (acute hepatitis B, hepatitis C, Leptospirosis), albeit most likely at sporadic level. Results strengthen the need for further water and sanitation interventions and are a stark reminder of the risk of other epidemics transmitted through similar routes in such settings, particularly dysentery and cholera. It also highlights the need to ensure clinical management capacity for potentially chronic conditions in this vulnerable population.

## Introduction

Since August 2017, an estimated 745,000 Rohingya—including more than 400,000 children below 18 years—have fled to Cox's Bazar district, Bangladesh from Rakhine state in Myanmar [1]. Pre-existing settlements, from earlier influxes rapidly expanded into large and dense mega-settlements. Today, an estimated total of 860,000 [2] stateless refugees live in 34 densely populated camps in Cox's Bazar district, in what has been described as one of the largest and fastest growing refugee crises in the world [3]. The rapid settlement of this vulnerable population into overcrowded camps, with inadequate provision of water, sanitation, or general hygiene standards, combined with the high population density are some of the factors that led to an increased risk of infectious disease outbreaks, compounded by vulnerability factors such as nutritional deficiencies and low pre-existing routine vaccination coverage [4].

An Early Warning, Alert and Response (EWAR) system was put in place by the World Health Organization (WHO) and the Bangladesh Ministry of Health and Family Welfare (MoHFW) in collaboration with multiple partners across the camps to help monitor and rapidly detect potential infectious diseases threats [5,6]. This allowed health facilities to report weekly case counts of different conditions/syndromes for which set thresholds were established which, when triggered, required investigation within 48 hours. In addition, EWAR health facilities immediately report unusual events of potential severity or any potential case or cluster requiring immediate investigation after initial verification of the event.

Acute Jaundice Syndrome (AJS) is not uncommon in refugee settings and can be caused by various infectious etiologies, such as acute viral hepatitis E, A, B, C, or more rarely leptospirosis, as well as non-infectious liver disease. The main differential diagnoses for these diseases and their transmission routes are summarised in S1 Appendix. Given the high population density and poor sanitation conditions usually found in such settings, hepatitis A and E can easily spread within a susceptible population. Of particular concern is Hepatitis E which can cause severe outcomes in pregnant women, with case fatality ratios (CFR) up to 10% reported in recent outbreaks [7–9] and which has been the leading cause of acute hepatitis in pregnancy in Myanmar [10]. In addition, Hepatitis E is less endemic, meaning that a lower proportion of adults tend to be immune, compared to hepatitis A, which classically affects mostly younger age groups in highly-endemic settings [7–9].

The prevalence of chronic hepatitis (B and C), vary among refugee populations [11]. The country of origin of patients has shown to be an important factor and Myanmar has a high reported incidence of acute and chronic liver disease, mostly caused by Hepatitis B virus (HBV) and Hepatitis C virus (HCV) [10,12]. The incidence of leptospirosis is not well documented in Bangladesh due to lack of diagnostic tests, though it has been reported in the region and the environmental factors are present to spread the infection [13]. Similarly, the incidence of leptospirosis in Myanmar has not been studied to date.

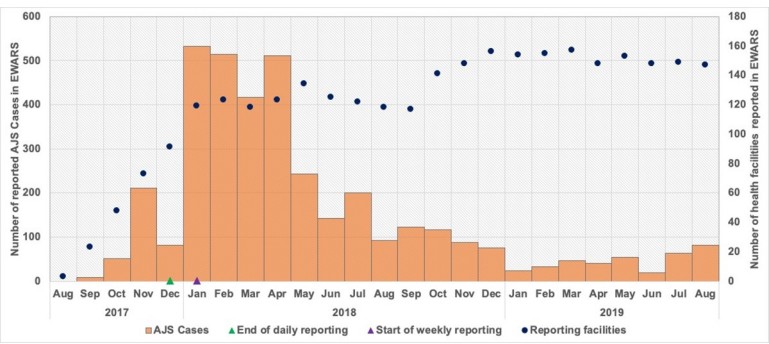

**Fig 1. Epidemiology curve of reported AJS cases against number of facilities reporting (monthly average) in EWAR system; and timing of changes to the surveillance system from daily to weekly.**

In January 2018, alerts for Acute Jaundice Syndrome (AJS) were triggered by the surveillance system due to a rapid increase in the reported weekly incidence (initially defined as twice the average of the last three weeks reported number of cases, from each facility). Between January and April 2018, a total of 1976 cases were reported through the system, with a weekly number of 77–147 cases (average 116 per week). The number of reporting health facilities remained stable (between 110–120 per week) throughout that period, and increased to 140 by the end of the year after reported case counts tailed off (Fig 1).

Following the alerts triggered by the surveillance system, and with over 1300 reported cases and 6 deaths since late 2017, a rapid field investigation was undertaken between 5-8th February 2018. Findings from these suggested multiple etiologies with initial laboratory results from 27 patient samples collected showing 37% (10/27) positive for HAV IgM, 7% (2/27) positive for HCV, 4% (1/27) positive for HEV, and the remaining 52% (14/27) negative for all Hepatitis markers and Leptospirosis IgM. The presence of multiple etiologies found in this initial investigation, especially HAV and HEV, prompted the simultaneous establishment of an enhanced surveillance strategy to ensure detailed information about each case could be gathered and analysed, combined with and exhaustive laboratory outbreak investigation to better understand the etiologies. Here, we describe the process and findings from this enhanced surveillance and outbreak investigation.

## Methods

### Establishment of enhanced surveillance

An enhanced surveillance for AJS including daily reporting of cases was implemented since the beginning of 2018 in Cox's Bazar, with the objective to identify more detailed epidemiological characteristics of reported cases. A digital case report form was developed and available on the EWAR system [6]. The form included information on detailed demographic characteristics, geographical origin of patients, clinical presentation, exposure factors (water and sanitation exposure, household environment) and laboratory results for any samples that were taken (S2 Appendix).

### Exhaustive laboratory outbreak investigation

A serological investigation was conducted for approximately one month between 28 February and 26 March 2018 based on exhaustive testing of all patients presenting at a subset of 18 healthcare facilities in the camps. The main purpose of testing was to better document the etiologies at play, assuming likely co-circulation of HAV and HEV as the main causative agents,

and other acute (or flare ups of chronic) HBV and HCV, potential cases of leptospirosis as well as acute hepatitis from non-infectious causes. The duration (four weeks) of the enhanced laboratory investigation was based on an opportunistic sampling exercise which was required to provide enough understanding of the key drivers of infection, based on a distribution of etiologies in an initial sample of 27 patients (collected between 5-8th February 2018), as well as pragmatic considerations of stock availability, transport, storage and sample collection and testing capacity.

Participating health facilities were proposed as those who reported >10 cases of AJS between January and February 2018 (week 1–8) in EWAR, and who self-identified as having the capacity to collect and transport samples.

Any person with acute onset of jaundice- with or without fever and absence of any known precipitating factors (EWARS Bangladesh Case Definition for Acute Jaundice Syndrome, see S3 Appendix) fulfilling the below criteria was invited to provide a blood sample:

1. presenting at one of the 18 participating facilities (Fig 2).

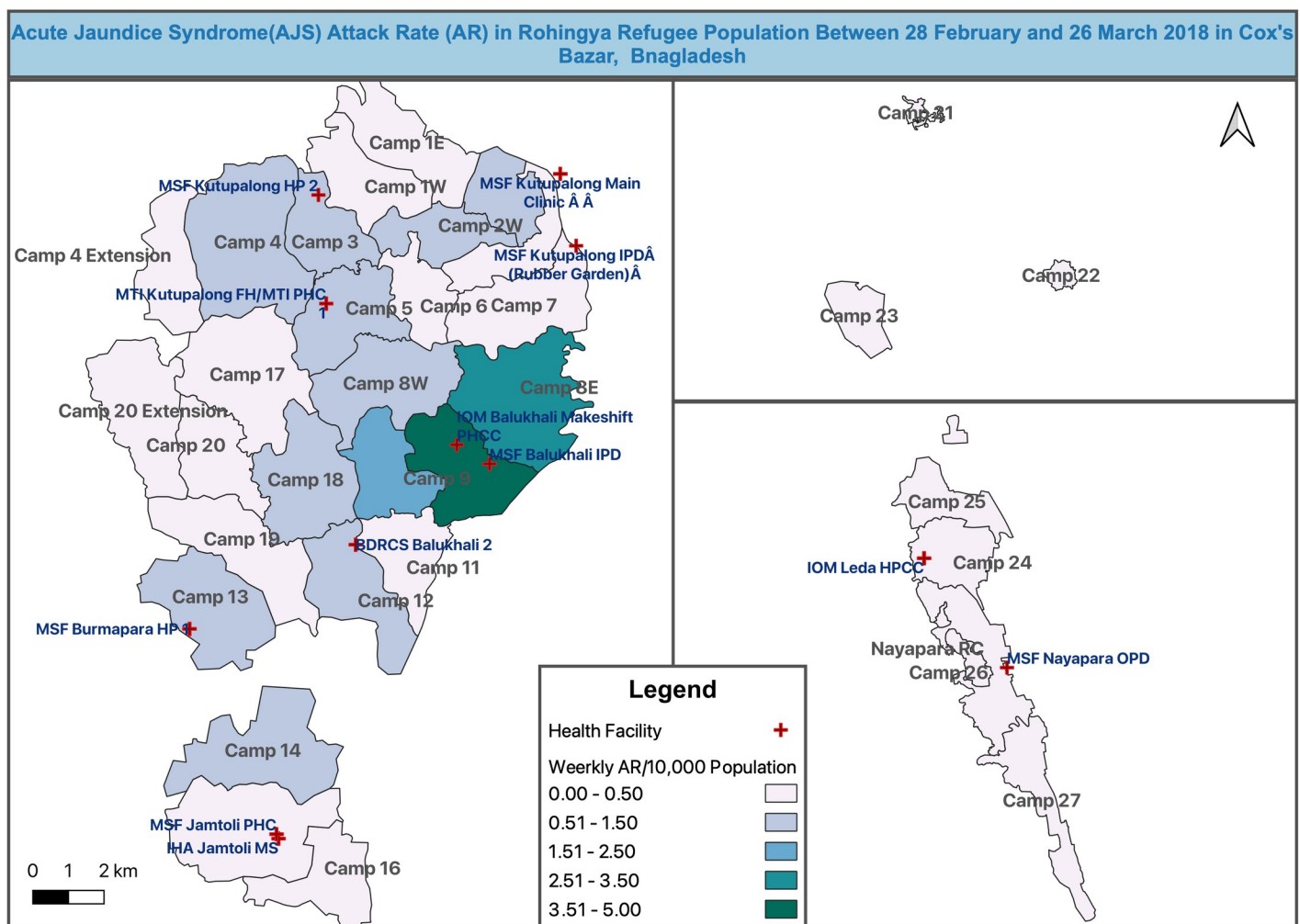

**Fig 2. Map of AJS attack rate (per 10,000 population) reported through enhanced epidemiological surveillance strategy between 28 February and 26 March 2018 in the Rohingya refugee camps and locations of reporting health facilities in Cox's Bazar, Bangladesh © 2021 by Md Khadimul Anam Mazhar is licensed under CC BY 4.0.**

2. with a completed Case Report Form (CRF), including demographic details, clinical details, family history, patient outcome

3. Having provided consent to having a sample taken and other information collected.

Participants were provided with information about the reason for the blood tests, and how their information would be handled and stored. Verbal consent was obtained by the medical team in charge at each of the 18 participating facilities. Participants who refused to provide consent, were excluded from sample collection.

## Laboratory procedures

Samples (venous blood) were collected in a pro-coagulation (red-top) tube and stored vertically maintaining a cold chain at 2–8˚C (for coagulation) before serum separation. Samples were labeled with unique IDs and transferred to a central facility (Médecins sans Frontières Kutupalong Clinic) for serum separation. Samples were later transported to the laboratory at the Institute of Epidemiology Disease Control and Research (IEDCR) in Dhaka for etiological analyses.

All samples were tested for HAV, HBV, HCV, HEV and leptospirosis using the diagnostic tests indicated in Table 1. Hepatitis testing was done using ELISA and Leptospira-IgM was tested using rapid immunochromatographic test (ICT).

## Statistical analyses

Descriptive analyses were conducted for all reported AJS cases followed by a post-hoc nested case control (case-case) analysis within the cohort of patients tested during the enhanced surveillance phase, to identify specific factors associated with hepatitis A, which was identified as the main cause for the AJS epidemic. In the analysis "Case" was classified as having Positive Hep A IgM result (HAV +ve cases) and "Control" was classified as having Negative Hep A IgM result (HAV -ve cases). Controls are not from the general population but from same cohort who presented with AJS symptoms and tested negative for Hepatitis A marker, irrespective of test results for other etiologies (Hepatitis B, C, E and Leptospirosis). Analysis was conducted to determine risk factors associated with predicted outcome (disease development) and effects of other independent variables (gender, age, age-groups, reported household transmission etc.) by measuring odds ratio with 95% confidence interval. All analysis was conducted using Stata 16.

## Environmental investigation

No specific investigation was undertaken during the enhanced surveillance phase for AJS, however findings from a large water quality surveillance exercise conducted shortly before the increased incidence of AJS alerts (18 September-14 November 2017) identified significant

**Table 1. Testing algorithm.**

| No | Disease/Pathogen | Diagnostic test to be performed |
|---|---|---|
| 1 | Hepatitis A Virus | HAV ELISA for IgM |
| 2 | Hepatitis B Virus | Hepatitis B surface antigen |
| 3 | Hepatitis C Virus | Anti-HCV antibodies |
| 4 | Hepatitis E Virus | Anti-HEV ELISA for IgM |
| 5 | Leptospirosis | Leptospira lateral flow for IgM |

contamination at water sources and household levels, underlining the presence of conditions for spread of water-borne disease (S4 Appendix).

### Ethical consideration

The data reported were collected as part of routine public health outbreak response activities. For this publication, no additional primary data collection took place other than the data required for outbreak response and surveillance. Therefore, no additional ethical approval was sought at the time of data collection. The formulation of this work was presented and discussed with the Civil Surgeon Cox's Bazar from the Ministry of Health & Family Welfare (MoHFW) responsible for overseeing the Rohingya refugee crisis. All individual case data was anonymized.

### Results

A total of 575 AJS cases were reported through weekly reporting with a weekly average of 115 cases ranging from 88 to 158 cases from 28 February to 26 March 2018 between week 8 to week 12 in 2018. Of these reported cases, 275 were reported in the enhanced surveillance system from whom blood samples were collected. Of these, 57% (156) were male and 68% (186) were children (below <18 years) (Table 2).

Camp-wise weekly attack rate (AR) per 10,000 population was calculated for the reported 275 AJS cases detected by the enhanced epidemiological surveillance strategy between 28 February and 26 March 2018. The weekly mean attack rate over the enhanced surveillance period (28 February to 26 March 2018) was 0.60 (95% CI: 0.28 to 0.92) per 10,000 population, across all the camps for the reported AJS cases. However, the AR varies among age-groups and was higher in younger age-groups (see Table 3). AR (per 10,000 population) for different age-groups is graphically presented in S5 Appendix.

All AJS cases presented with jaundice, as per reporting definition, combined with other symptoms. In total 67% (183/275) presented with fever, 56% (155/275) with nausea, 41% (113/275) with abdominal pain, 25% (70/275) with vomiting, 21% (57/275) with fatigue, 14% (38/275) with itching, 13% (37/275) with joint pain, 9% (27/275) with loss of appetite, 9% (26/275) with dark urine, 3% (7/275) with bleeding and 2% (6/275) with convulsion (Fig 3).

Out of the 275 samples, 206 were positive for at least one of the agents tested. The laboratory results confirmed multiple etiologies including 154 (56%) samples tested positive for Hepatitis A, 1 (0.4%) positive for Hepatitis E, 36 (13%) positive for Hepatitis B, 25 (9%) positive for Hepatitis C, and 14 (5%) positive for Leptospirosis.

Among all specimens tested 24 (9%) showed evidence of co-infections with multiple etiologies. Of these, 4% (11/275) had HAV-HBV co-infection, 0.7% (2/275) had HAV-HCV co-infection, 0.4% (1/275) had HAV-HEV co-infection, 0.4% (1/275) had HAV-LEPT co-infection; 0.7% (2/275) had HBV-HCV co-infection, 0.4% (1/275) had HBV-LEPT co-infection and 0.4% (1/275) had HCV-LEPT co-infection. The remaining 69 (25%) samples tested negative for all tests.

**Table 2. Gender and age-group distribution of reported AJS cases.**

| Age Group of Reported Cases | Male | Female | Total |
|---|---|---|---|
| 0 to 4 years | 45 (16%) | 22 (8%) | 67 (24%) |
| 5 to 9 Years | 34 (12%) | 25 (9%) | 59 (21%) |
| 10 to 17 Years | 34 (12.3%) | 26 (9.4%) | 60 (21.7%) |
| 18 years or more | 43 (15.6%) | 46 (16.7%) | 89 (32.3%) |
| **Total** | **156 (56.7%)** | **119 (43.3%)** | **275 (100%)** |

**Table 3. Attack rate of reported AJS cases by age-group** * **and gender among all 34 camps between 28 February and 26 March 2018 in Cox's Bazar, Bangladesh.**

| | Weekly AR/10,000 population (95% CI) |
|---|---|
| Overall | 0.60 (0.28 to 0.92) |
| **By age-group** | |
| 0–4 years | 0.80 (0.34 to 1.26) |
| 5–11 years | 0.73 (0.36 to 1.10) |
| 12–17 years | 0.71 (0.31 to 1.11) |
| 18+ years | 0.43 (0.15 to 0.92) |
| **By gender** | |
| Female | 0.49 (0.21 to 0.77) |
| Male | 0.71 (0.33 to 1.09) |

* Population data from UNHCR Bangladesh Operational Update, 1–15 August 2018, was used to calculate the AJS attack rate [14].

Descriptive age-group analysis showed that among all Hepatitis A positives cases (154), 38% (58/154) were aged 0–4 years, 34% (3252/154) were aged 5–9 years, 21% (32/154) were aged 10–17 years and the remaining 8% (12/154) were adults (18 years or above). The only case found to be positive for Hepatitis E was aged 5–9 years. Half (18/36) of the Hepatitis B positives case were reported among younger age-groups (<18 years), whereas for HCV only 8% (2/25) were reported in younger age-groups (<18 years). Among all (14) leptospirosis cases there were male predominance (64%) and all cases were in older children (29% in 10 to 17 years) and in adult (71% in 18 years or above) age groups (Fig 4).

Among all AJS cases, 8% (23/ 275) reported household exposure, of which 13 (56%) were female and 10 (44%) were male which represents 11% (13/113) of all reported female and 6.4% (10/156) all reported male AJS cases. Of these 23 cases reported with household exposure, only 11 females and 5 males tested positive for Hepatitis A.

Additional analyses were performed to examine the association between hepatitis A sero-positivity with age, gender and other risk factors. Univariate regression analysis showed that the odds of developing hepatitis A was lower in older age groups (10–17 and 18+ years) compared to younger age-group (0–4 years) with a p-value of <0.001. (Table 4). Univariate

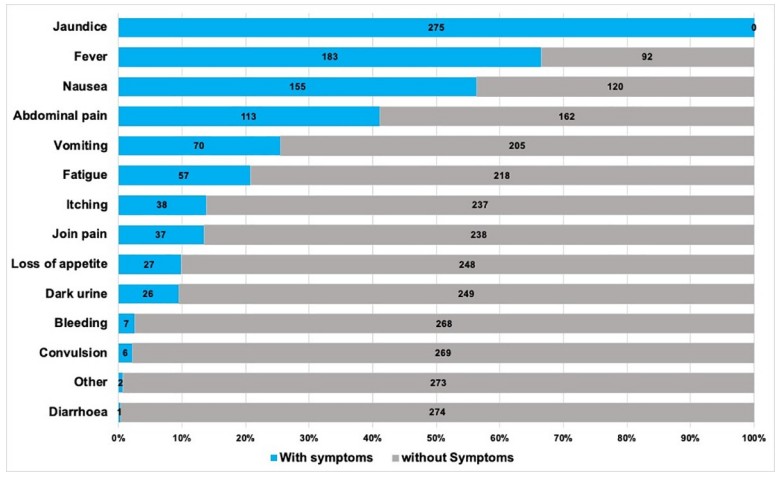

**Fig 3. Presenting symptoms of reported AJS cases.**

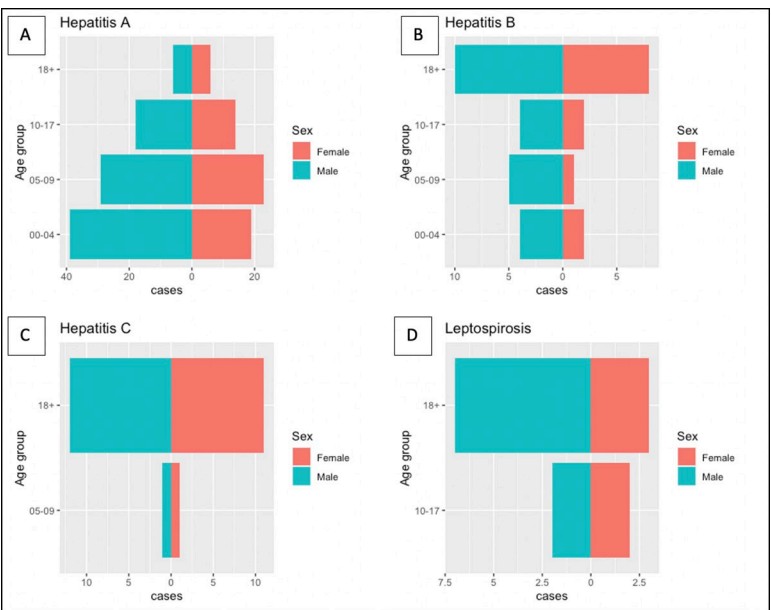

**Fig 4. Age and sex distribution of AJS cases with their seropositivity collected during exhaustive sampling, 28 February– 26 March 2018, Cox's Bazar, Bangladesh.** (A) Hepatitis A seropositivity. (B) Hepatitis B seropositivity. (C) Hepatitis C seropositivity. (D) Leptospirosis seropositivity.

analysis also showed no association between hepatitis A seropositivity and gender (p-value 0.256) (Table 4).

We explored whether presenting symptoms among AJS cases were associated with hepatitis A seropositivity. Univariate analysis showed that HAV was positively associated with fever (OR 2.25; 95% CI: 1.35 to 3.76); dark urine (OR 3.63; 95% CI: 1.32 to 9.94); loss of appetite (OR 3.83; 95% CI: 1.40 to 10.45) and vomiting (OR 1.85; 95% CI: 1.05 to 3.27), and negatively associated with joint pain (OR 0.42; 95% CI 0.21–0.86) (Table 4). Etiological risk factor data collected from the cases indicate different sources were used for drinking water including tube wells, communal tap, water supply trucks. However, univariate regression analysis showed no association between reported drinking water source type and hepatitis A seropositivity (Table 4 and Fig 5).

Additional nested case control (case-case study) analyses showed no association between hepatitis A seropositivity and a reported case of AJS in the household (p-value 1.77). However, sub-group analysis showed that female HAV cases were more likely to have reported house-hold transmission (for definition see S6 Appendix) than males, with a OR of 5.93 (95% CI:1.25 to 28.1; p-value: 0.025). The univariate regression analysis among age-groups showed that, compared to younger age groups (0–4 years; 5–9 years and 10–17 years), adult HAV cases (18 years or above) were more likely to have reported household transmission with an OR of 18.75 (95% CI: 2.96 to 118.92; p-value: 0.002) (see Table 5).

## Discussion

The investigation into an outbreak of acute jaundice syndrome in the Rohingya refugee camps revealed a hepatitis A epidemic and multiple other infectious etiologies contributing to AJS (acute HBV, HCV, leptospirosis). Data from enhanced epidemiological surveillance suggests that at least 154 were infected (56% of the reported AJS cases) with HAV between February and March 2018 which was the most common underlying cause of AJS among the tested cases,

**Table 4. Association of hepatitis A seropositivity with age-group, gender, presenting symptoms and drinking water sources.**

|  | OR (95% CI) | P-value |
|---|---|---|
| **Age-group** | | |
| 0–4 years (n = 67) | 1.0* | - |
| 5–9 years (n = 59) | 1.15 (0.40 to 3.13) | 0.79 |
| 10–17 years (n = 60) | 0.18 (0.07 to 0.42) | < 0.001 |
| 18+ years (n = 89) | 0.02 (0.01 to 0.06) | < 0.001 |
| **Gender** | | |
| Female (n = 119) | 1.0* | - |
| Male (n = 156) | 1.32 (0.82 to 2.14) | 0.256 |
| **Presenting symptoms#** | | |
| Fever (n = 183) | 2.25 (1.35 to 3.76) | 0.002 |
| Nausea (n = 155) | 1.19 (0.76 to 1.93) | 0.479 |
| Abdominal pain (n = 113) | 0.81 (0.50 to 1.31) | 0.385 |
| Vomiting (n = 70) | 1.85 (1.05 to 3.27) | 0.034 |
| Fatigue (n = 57) | 0.83 (0.46 to 1.50) | 0.542 |
| Itching (n = 38) | 1.23 (0.61 to 2.47) | 0.563 |
| Joint pain (n = 37) | 0.42 (0.21 to 0.86) | 0.018 |
| Loss of Appetite (n = 27) | 3.83 (1.40 to 10.45) | 0.009 |
| Dark Urine (n = 26) | 3.63 (1.32 to 9.94) | 0.012 |
| Bleeding (n = 7) | 1.04 (0.23 to 4.74) | 0.960 |
| Convulsion (n = 6) | 0.38 (0.07 to 2.12) | 0.271 |
| **Drinking water sources^** | | |
| Tube-well (n = 170) | 1.27 (0.78 to 2.06) | 0.342 |
| Tap water (n = 45) | 1.22 (0.63 to 2.33) | 0.555 |
| Unknown sources (n = 58) | 0.56 (0.32 to 1.01) | 0.055 |

*Baseline category

#Diarrhoea (n = 1) and other (n = 2) symptoms were omitted from the summary table

^ water truck (n = 2) was omitted from the summary table.

and driving the epidemic dynamics, followed by Hepatitis B and hepatitis C which likely contributed to background levels of AJS in the camps Hepatitis E was observed in only one of the 275 samples.

The results highlight the importance of early warning, alert and response (EWAR) system in humanitarian contexts, as well as the need for such surveillance processes to be versatile and easily adaptable to respond to new outbreaks as they arise. Here, following the identification of an AJS outbreak through EWAR, a short-term enhanced surveillance system was put in place to investigate the potential causative agents and related risk factors linked to them, in order to guide outbreak response.

Since sampling was conducted at selected sites and during a short period of time, results are likely to have been exposed to selection bias. The results nevertheless provided a good overview of the causative etiologies of AJS in the camps.

Sub-standard water and sanitation conditions and lack of hygiene practices in the camps were observed at the time of data collection as indicated by the findings from the water quality surveillance (S Appendix), coupled with crowded living conditions are fertile grounds for enteric and waterborne pathogens, including hepatitis A. Hepatitis A and E are most commonly transmitted via the fecal-oral route [15], and Consumption of contaminated food and

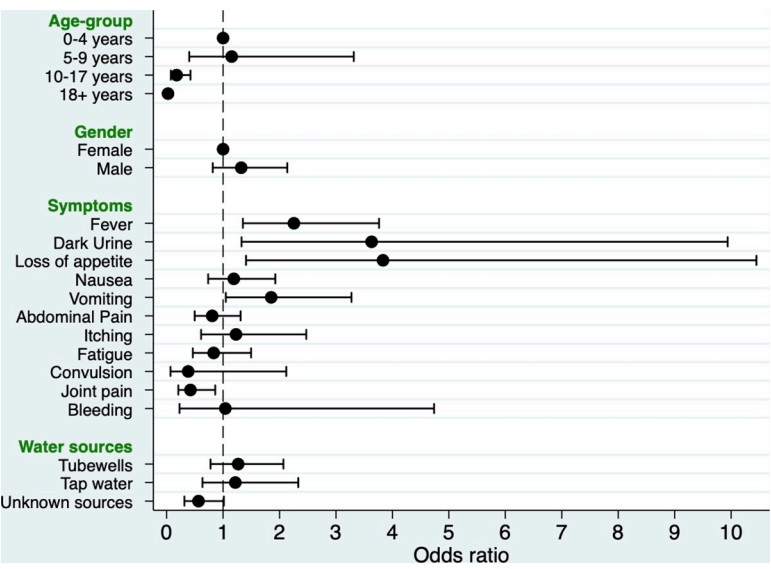

**Fig 5. Odds ratio for the association of hepatitis A seropositivity with age-group, gender, presenting symptoms and drinking water sources.**

water is the leading cause of HAV and HEV outbreaks in refugee camp settings [7]. In high incidence settings, hepatitis A infection tend to mostly affect children, due to existing immunity in older age groups, as corroborated by our results. One of the concerns at the start of the epidemic was a Hepatitis E outbreak, which may cause fetal loss and high maternal mortality, and which has often been associated with large water borne outbreaks [16]. HEV is also one of the most common causes of acute viral hepatitis in this region and is common in rural areas in

**Table 5. Odds ratio for the association between household transmission and hepatitis A seropositivity by gender.**

| AJS case in the household** | | OR (95% CI) | P-value |
|---|---|---|---|
| Overall | Hepatitis A negative (n = 121) | 1.0* | - |
| | Hepatitis A positive (n = 154) | 1.88 (0.75 to 4.75) | 0.177 |
| **By Gender** | | | |
| Female | Hepatitis A negative (n = 57) | 1.0* | - |
| | Hepatitis A positive (n = 62) | 5.93 (1.25 to 28.06) | 0.025 |
| Male | Hepatitis A negative (n = 64) | 1.0* | - |
| | Hepatitis A positive (n = 92) | 0.68 (0.19 to 2.45) | 0.553 |
| **By age-groups** | | | |
| 0–4 years | Hepatitis A negative (n = 9) | 1.0* | - |
| | Hepatitis A positive (n = 58) | 0.15 (0.03 to 0.83) | 0.029 |
| 5–9 years | Hepatitis A negative (n = 7) | 1.0* | - |
| | Hepatitis A positive (n = 52) | 1.0 (omitted) | - |
| 10–17 years | Hepatitis A negative (n = 28) | 1.0* | - |
| | Hepatitis A positive (n = 32) | 0.87 (0.11 to 6.59) | 0.890 |
| 18+ years | Hepatitis A negative (n = 77) | 1.0* | - |
| | Hepatitis A positive (n = 12) | 18.75 (2.96 to 118.92) | 0.002 |

*Baseline category

**In the 8 weeks preceding symptom onset.

Bangladesh [17,18]. While the investigation did not point towards extensive HEV circulation, presence of the virus and the similarity of transmission routes with Hepatitis A called for particular attention to be given to pregnant women, who are at high risk of complications if affected by HEV within the third trimester, and continued vigilance of any flare up of AJS in the camps in the future.

Our investigation showed that, among adults, the risk of HAV positivity was higher for individuals who reported previous AJS cases in their household, within 8 weeks, and females in particular had a higher risk. This may reflect differential risks of infection within the household setting, potentially linked to care giving roles.

While Hepatitis B and C were not drivers of the AJS epidemic, the detection of significant number of infections requires attention and likely reflects high prevalence of chronic infections. l. One small study has shown significant prevalence of HBV and HCV among the Rohingya population in Bangladesh [15] and warrants further serological investigation in particular given limited capacity for clinical management within the district. Hepatitis B vaccination is part of the routine EPI programmes in the camps, and our results highlight the need to ensure adequate coverage, as well as a zero dose to new-borns to protect from spread of the infection, with close monitoring of identified cases. There is also need to maintain routine immunization programmes, to improve vaccine coverage of Hepatitis B among women of reproductive age (WRA) and under-five children both in the camps and the surrounding host community [4]. This might, in turn, reduce the perinatal transmission of hepatitis viruses [19].

Presence of leptospirosis has been a surprising but not unexpected finding. Laboratory testing for leptospirosis is not commonly practiced in Cox's Bazar. The camps have associated environmental factors like a temperate climate and floods due to heavy rainfalls and recreational activities including swimming are common, so these may spread the infection [20]. Environmental exposures are likely higher in male and also in older age groups and might contribute to developing the disease.

For conducting this rapid investigation in the Rohingya camps, one of the biggest challenges was to ensure a minimum standard for sample collection. At the time of the investigation, very few facilities were using rapid tests to confirm clinical diagnosis for Hepatitis A and Hepatitis E. To improve the diagnostic capacities and appropriate clinical management rapid diagnostic tests should be available and tests should be done routinely. Sample collection for blood and stool have since been incorporated into the Minimum Service Package required for health facilities in the camps.

AJS has been shown to be a regular public health problem in refugee settings, and in the Rohingya population in Cox's Bazar district in particular. The characterization of the causative etiology is important to tailor appropriate preventive and responsive measures. Our results highlight the need to continue efforts to improve the long-term quality of water and sanitation in the camps, as well as for an improved detection and clinical management of Hepatitis B and C.

## Supporting information

**S1 Appendix. Etiologies of acute jaundice syndrome.**
(PDF)

**S2 Appendix. Acute jaundice syndrome case report form.**
(PDF)

**S3 Appendix. Acute jaundice syndrome—case definition.**
(PDF)

**S4 Appendix. Water quality testing results in refugee settlements from 18 September to 14 November 2017, Cox's Bazar, Bangladesh [21].**
(PDF)

**S5 Appendix. Map of AJS attack rate (per 10,000 population) by age-groups reported during enhanced epidemiological surveillance strategy (between 28 February and 26 March 2018) in the Rohingya refugee camps in Cox's Bazar, Bangladesh © 2021 by Md Khadimul Anam Mazhar is licensed under CC BY 4.0.**
(TIF)

**S6 Appendix. Reported AJS household transmission–definition.**
(PDF)

## Acknowledgments

We would like to thank the Civil Surgeon of Cox's Bazar District of Ministry of Health and Family Welfare (MoHFW) for his full support and guidance during the outbreak response. We would also like to thank colleagues from WHO Operations Support and Logistics team for their relentless support throughout the outbreak response. We are grateful to Cox's Bazar Medical College Hospital for their overall support, guidance and supervision in laboratory investigation. We would also like to acknowledge the Global Outbreak, Alert and Response Network (GOARN) operational support team for the assistance with deployments staff supporting from all around the globe. Special thanks to all the organizations working for Rohingya refugee response under the Health Sector for their support to this investigation. Lastly, we would like to thank the Rohingya refugees for their resilience and strength in ongoing difficult times.

## Author Contributions

**Conceptualization:** Md Khadimul Anam Mazhar, Flavio Finger, Egmond Samir Evers, Anna Kuehne, Melissa Ivey, Tahmina Shirin, Kate White, Amrish Baidjoe, Olivier le Polain de Waroux.

**Data curation:** Md Khadimul Anam Mazhar, Flavio Finger, Anna Kuehne, Melissa Ivey, Tahmina Shirin.

**Formal analysis:** Md Khadimul Anam Mazhar, Flavio Finger.

**Investigation:** Md Khadimul Anam Mazhar, Flavio Finger, Anna Kuehne, Melissa Ivey, Francis Yesurajan, Tahmina Shirin, Nurul Ajim, Ahammadul Kabir, Kate White, Olivier le Polain de Waroux.

**Methodology:** Md Khadimul Anam Mazhar, Flavio Finger, Anna Kuehne, Melissa Ivey, Francis Yesurajan, Kate White, Olivier le Polain de Waroux.

**Project administration:** Md Khadimul Anam Mazhar, Egmond Samir Evers, Melissa Ivey, Jennie Musto, Kate White, Olivier le Polain de Waroux.

**Resources:** Md Khadimul Anam Mazhar, Egmond Samir Evers, Tahmina Shirin, Jennie Musto.

**Software:** Md Khadimul Anam Mazhar.

**Supervision:** Anna Kuehne, Tahmina Shirin, Amrish Baidjoe, Olivier le Polain de Waroux.

**Validation:** Amrish Baidjoe, Olivier le Polain de Waroux.

**Visualization:** Md Khadimul Anam Mazhar.

**Writing – original draft:** Md Khadimul Anam Mazhar, Flavio Finger, Egmond Samir Evers.

**Writing – review & editing:** Md Khadimul Anam Mazhar, Flavio Finger, Egmond Samir Evers, Anna Kuehne, Francis Yesurajan, Tahmina Shirin, Nurul Ajim, Ahammadul Kabir, Jennie Musto, Kate White, Amrish Baidjoe, Olivier le Polain de Waroux.

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
