## [Decision Letter · Decision Letter 0]

4 Dec 2020

PONE-D-20-32697

Hepatitis A outbreak among the Rohingya refugees in Cox’s Bazar, Bangladesh:  findings from enhanced epidemiological surveillance

PLOS ONE

Dear Dr. Mazhar,

Thank you for submitting your manuscript to PLOS ONE. After careful consideration, we feel that it has merit but does not fully meet PLOS ONE’s publication criteria as it currently stands. Therefore, we invite you to submit a revised version of the manuscript that addresses the points raised during the review process.

We look forward to receiving your revised manuscript.

Kind regards,

Hans Tillmann

Academic Editor

PLOS ONE

Journal Requirements:

2. Thank you for including your ethics statement provided within your manuscript:

"The data reported were collected as public health outbreak response activities. For this 180publication, no additional primary data collection took place. However, theanalysis was done with the support of and in consultation withthe Civil Surgeon, Cox’s Bazar, Bangladesh. All individual case data was anonymized."

3. Please provide additional details regarding participant consent.

In the ethics statement in the Methods and online submission information, please ensure that you have specified (i) whether consent was informed and (ii) what type you obtained (for instance, written or verbal, and if verbal, how it was documented and witnessed). If your study included minors, state whether you obtained consent from parents or guardians. If the need for consent was waived by the ethics committee, please include this information.

'The funders had no role in study design, data collection and analysis, decision to publish, or preparation of the manuscript'

6. Please include a separate caption for each figure in your manuscript.

7. Please ensure that you refer to Figure 2 in your text as, if accepted, production will need this reference to link the reader to the figure.

8. We note that Supporting Information Appendix 3 in your submission contains map images which may be copyrighted.

We require you to either (a) present written permission from the copyright holder to publish these figure specifically under the CC BY 4.0 license, or (b) remove the figure from your submission:

a. You may seek permission from the original copyright holder of Supporting Information Appendix 3 to publish the content specifically under the CC BY 4.0 license. 

b. If you are unable to obtain permission from the original copyright holder to publish this figure under the CC BY 4.0 license or if the copyright holder’s requirements are incompatible with the CC BY 4.0 license, please either i) remove the figure or ii) supply a replacement figure that complies with the CC BY 4.0 license. Please check copyright information on all replacement figures and update the figure caption with source information. If applicable, please specify in the figure caption text when a figure is similar but not identical to the original image and is therefore for illustrative purposes only.

9. Please include captions for your Supporting Information files at the end of your manuscript, and update any in-text citations to match accordingly. Please see our Supporting Information guidelines for more information: http://journals.plos.org/plosone/s/supporting-information

Reviewers' comments:

It is an article that describes an outbreak of viral hepatitis A associated with poor sanitary conditions in the refugee population. The presence of other etiologies, such as HBV, may be due to the fact that they already had a previous infection, considering that they are endemic areas; however, a comment could be added on the horizontal transmission of HBV in conglomerates of people. A map of the refugee location could be included.

It is important to improve bibliographic references in general. The data in the tables and graphs are repeated and could be described in the text. If it is published it could be like a short communication.

**Major compulsory revisions**

**All the comments on the paper need to be addressed as follows:**

<h5>**1.     As the refugees migration took place in Summer of 2017 means May, June as starting point and as per epi curve the data of AJS cases showed approx.. more than 3500 cases and 6 deaths but 6 deaths were not shown and not mentioned in this articles, please include**</h5>

<h5>**2.     Location map is very important to look for clustering of cases in camps. As per Epi-curve, there is gradually increase of no. of health units under surveillance (EWARS) approx. 160 units but data had been shown to only 18 health units for 4 weeks, most probably because of more cases in Feb to March 2018 in those units, but we missed which parts of camp are more affected and we can explain better if we put map of location and camps in COX’s bazar.**</h5>

<h5>**3.     We cannot comment odd ratio without table and percentage: Naked odds ratios give no idea about the extent of this risk factor.  It’s always better to include the percent of cases and percent of controls exposed.  One could get an OR of 5 with only 20% exposed.  **</h5>

**Please send word documents more comments on track change mode.**

**Minor essential revisions**

**The paper could be improved by thorough editing and formatting.**

---

## [Author Response · Author response to Decision Letter 0]

2 Feb 2021

Response to Reviewers:

Journal Requirements:

A. When submitting your revision, we need you to address these additional requirements.

We thank the reviewers for drawing attention to meet PLOS One’s style requirements. We have made necessary amendments in the manuscript and file naming during our re-submission. 

2. Thank you for including your ethics statement provided within your manuscript:

"The data reported were collected as public health outbreak response activities. For this 180publication, no additional primary data collection took place. However, theanalysis was done with the support of and in consultation withthe Civil Surgeon, Cox’s Bazar, Bangladesh. All individual case data was anonymized."

We thank the reviewers for drawing attention to ethics statement. The data reported were collected as part of routine public health outbreak response activities. For this publication, no additional primary data collection took place other than the data required for outbreak response and surveillance. The formulation of this work was presented, reviewed and approved by the review committee for “Operational research during Rohingya Refugee crisis” is led by the Ministry of Health & Family Welfare (MoHFW), Bangladesh. Written approval to publish this manuscript in a journal was obtained from the Ministry of Health & Family Welfare (MoHFW), Bangladesh responsible for the oversight the response of the refugee crisis in Cox’s Bazar, Bangladesh. We have updated the ethics statement as follows as follows (lines 190 to 197): 

“The data reported were collected as part of routine public health outbreak response activities. For this publication, no additional primary data collection took place other than the data required for outbreak response and surveillance. Therefore, no additional ethical approval was sought at the time of data collection. The formulation of this work was presented and discussed with the Civil Surgeon Cox’s Bazar from the Ministry of Health & Family Welfare (MoHFW) responsible for overseeing the Rohingya refugee crisis. All individual case data was anonymized.”

3. Please provide additional details regarding participant consent.

In the ethics statement in the Methods and online submission information, please ensure that you have specified (i) whether consent was informed and (ii) what type you obtained (for instance, written or verbal, and if verbal, how it was documented and witnessed). If your study included minors, state whether you obtained consent from parents or guardians. If the need for consent was waived by the ethics committee, please include this information.

We thank the reviewers for drawing attention regarding participant consent. We have updated the text regarding participants consent (lines 155 to 158):

“Participants were provided with information about the reason for the blood tests, and how their information would be handled and stored. Verbal consent was obtained by the medical team in charge at each of the 18 participating facilities. Participants who refused to provide consent, were excluded from sample collection.”

We thank the reviewers for drawing attention to data availability statement. The data used to develop the manuscript related to the Rohingya refugee population living in Cox’s Bazar and are owned by Ministry of Health & Family Welfare (MoHFW). Due to the political sensitivity of the subject matter, there is restriction on sharing information on Rohingya refugee population. A written approval was obtained from the MoHFW to use this anonymised data set only for the purpose of sharing the results of this outbreak investigation and in the context of this manuscript. Any further request to access the data has to be approved by the MoHFW responsible for overseeing the Rohingya refugee response. The contact details are given below – 

– Civil Surgeon Office, Ministry of Health & Family Welfare,

– Kolatoli Road, Cox’s Bazar, Bangladesh

– Email: coxsbazar@cs.dghs.gov.bd

'The funders had no role in study design, data collection and analysis, decision to publish, or preparation of the manuscript'

a. Please clarify the sources of funding (financial or material support) for your study. List the grants or organizations that supported your study, including funding received from your institution.

d. If you did not receive any funding for this study, please state: “The authors received no specific funding for this work.”

We thank the reviewers for drawing attention to financial disclosure. We have made following changes in our manuscript (lines: 366 to 373) – 

“The authors received no specific funding for this work. The UK Public Health Rapid Support Team is funded by UK Aid from the Department of Health and Social Care and is jointly run by Public Health England and the London School of Hygiene & Tropical Medicine. The views expressed in this publication are those of the authors and not necessarily those of the National Health Service, the National Institute for Health Research or the Department of Health and Social Care. The funders had no role in study design, data collection and analysis, decision to publish, or preparation of the manuscript.” 

6. Please include a separate caption for each figure in your manuscript.

We thank the reviewers for drawing attention to this. We have made changes in to the manuscript accordingly. 

7. Please ensure that you refer to Figure 2 in your text as, if accepted, production will need this reference to link the reader to the figure.

We thank the reviewers for drawing attention to this. We have made changes in to the manuscript accordingly (lines: 104). 

8. We note that Supporting Information Appendix 3 in your submission contains map images which may be copyrighted.

We require you to either (a) present written permission from the copyright holder to publish these figure specifically under the CC BY 4.0 license, or (b) remove the figure from your submission:

a. You may seek permission from the original copyright holder of Supporting Information Appendix 3 to publish the content specifically under the CC BY 4.0 license. 

b. If you are unable to obtain permission from the original copyright holder to publish this figure under the CC BY 4.0 license or if the copyright holder’s requirements are incompatible with the CC BY 4.0 license, please either i) remove the figure or ii) supply a replacement figure that complies with the CC BY 4.0 license. Please check copyright information on all replacement figures and update the figure caption with source information. If applicable, please specify in the figure caption text when a figure is similar but not identical to the original image and is therefore for illustrative purposes only.

We thank the reviewers for drawing attention to the copyright issues regarding use of maps in the manuscript. The corresponding author is solely the copyright holder of the maps used in the manuscript. Additional copyright supporting document is also provided with the revised submission. 

9. Please include captions for your Supporting Information files at the end of your manuscript, and update any in-text citations to match accordingly. Please see our Supporting Information guidelines for more information: http://journals.plos.org/plosone/s/supporting-information

We thank the reviewers for drawing attention to this. We have made changes in to the manuscript. We added the segment of supporting information in the end of the manuscript (lines: 435 to 443)

B. Reviewers' comments:

It is an article that describes an outbreak of viral hepatitis A associated with poor sanitary conditions in the refugee population. The presence of other etiologies, such as HBV, may be due to the fact that they already had a previous infection, considering that they are endemic areas; however, a comment could be added on the horizontal transmission of HBV in conglomerates of people. A map of the refugee location could be included.

It is important to improve bibliographic references in general. The data in the tables and graphs are repeated and could be described in the text. If it is published it could be like a short communication.

We thank the reviewers for their valuable comments. We have added a comment on the horizontal transmission of HBV in the manuscript (line 330 to 333):

“Presence of high number of Hepatitis B in the younger age-groups indicated the need for improving immunization coverage of Hepatitis B among women of reproductive age (WRA) and among all newborns in the camps and the host community by strengthening routine immunization (4). This might in turn reduce the perinatal transmission of hepatitis viruses (18).”

We thank the reviewers for their suggestion to add a map of refugee location. We added an additional map showing the location of refugee population density in the camps titled – “Fig 1. Camp-wise distribution of the Rohingya refugee population located in Cox’s Bazar, Bangladesh” .

“ Fig 1. Camp-wise distribution of the Rohingya refugee population located in Cox’s Bazar, Bangladesh

 ”

We have also made changes according to the reviewer’s suggestion on bibliographic reference in general. 

Major compulsory revisions

All the comments on the paper need to be addressed as follows:

1. As the refugees migration took place in Summer of 2017 means May, June as starting point and as per epi curve the data of AJS cases showed approx. more than 3500 cases and 6 deaths but 6 deaths were not shown and not mentioned in this article, please include

 We thank the reviewers for their suggestion and added deaths in the epi curve titled – “Fig 2. Epidemiology curve of reported AJS cases against number of facilities reporting (monthly average) in EWAR system; and timing of changes to the surveillance system from daily to weekly”:

“Fig 2. Epidemiology curve of reported AJS cases against number of facilities reporting (monthly average) in EWAR system; and timing of changes to the surveillance system from daily to weekly

 ”

2. Location map is very important to look for clustering of cases in camps. As per Epi-curve, there is gradually increase of no. of health units under surveillance (EWARS) approx. 160 units but data had been shown to only 18 health units for 4 weeks, most probably because of more cases in Feb to March 2018 in those units, but we missed which parts of camp are more affected and we can explain better if we put map of location and camps in COX’s bazar.

We thank the reviewers for their suggestion, however due to resource constraints and lack of laboratory/diagnostic unit, it wasn’t possible to include all health units reporting to our surveillance network. We had to select only a handful of primary care units with diagnostic/laboratory unit (for collecting and processing blood samples).

3. We cannot comment odd ratio without table and percentage: Naked odds ratios give no idea about the extent of this risk factor. It’s always better to include the percent of cases and percent of controls exposed. One could get an OR of 5 with only 20% exposed. 

We thank the reviewers for their comments and we made necessary changes in to the manuscript. 

Please send word documents more comments on track change mode.

Minor essential revisions

The paper could be improved by thorough editing and formatting.

We thank the reviewers for their feedback. We have tried to improve our manuscript in our re-revised version.

---

## [Decision Letter · Decision Letter 1]

2 Mar 2021

PONE-D-20-32697R1

Hepatitis A outbreak among the Rohingya refugees in Cox’s Bazar, Bangladesh:  findings from enhanced epidemiological surveillance

PLOS ONE

Dear Dr. Mazhar,

Thank you for submitting your manuscript to PLOS ONE. After careful consideration, we feel that it has merit but does not fully meet PLOS ONE’s publication criteria as it currently stands. Therefore, we invite you to submit a revised version of the manuscript that addresses the points raised during the review process.

ACADEMIC EDITOR:

sorry I missed it in the initial submission, but can you explain the fatalities/death demonstrated in figure 1, and what is the purpuse of showing death, while you do not mention death otherwise in manuscript.Another good point by one reviewed, also missed in first go-around, the title may best include jaundice as starting point, and that HAV was found as dominant culprit.Also, did you had an HCV IgM test available? I am not aware that such test exist.I believe no one was HBV anti-HBc IgM positive?

We look forward to receiving your revised manuscript.

Kind regards,

Hans Tillmann

Academic Editor

PLOS ONE

Reviewers' comments:

Reviewer's Responses to Questions

**Comments to the Author**

1. If the authors have adequately addressed your comments raised in a previous round of review and you feel that this manuscript is now acceptable for publication, you may indicate that here to bypass the “Comments to the Author” section, enter your conflict of interest statement in the “Confidential to Editor” section, and submit your "Accept" recommendation.

Reviewer #1: All comments have been addressed

2. Is the manuscript technically sound, and do the data support the conclusions?

Reviewer #1: Yes

3. Has the statistical analysis been performed appropriately and rigorously? 

Reviewer #1: N/A

4. Have the authors made all data underlying the findings in their manuscript fully available?

Reviewer #1: Yes

5. Is the manuscript presented in an intelligible fashion and written in standard English?

Reviewer #1: Yes

6. Review Comments to the Author

Reviewer #1: 1 As per topic heading is concerned, “Hepatitis A outbreak among the Rohingya refugees in Cox’s Bazar, Bangladesh:findings from enhanced epidemiological surveillance”; it’s better to keep “Jaundice outbreak among the Rohingya refugees in Cox’s Bazar, Bangladesh: findings from enhanced epidemiological surveillance”- there were cases multifactorial etiologies found after investigation, so in my opinion Jaundice outbreak covers all details.

2 Regarding Hep B cases found during surveillance, warrants immunization Hep- B drive in all age group healthy individuals in 34 camps as well as zero doses to new borne ASAP to protect from spread and close monitoring to cases.

3 In last, I feel for better representation of data, descriptive data should be kept separate and analytical data should be separate.

7. PLOS authors have the option to publish the peer review history of their article (what does this mean?). If published, this will include your full peer review and any attached files.

Reviewer #1: **Yes: **Tripurari Kumar

---

## [Author Response · Author response to Decision Letter 1]

3 Apr 2021

Response to Reviewers:

1. ACADEMIC EDITOR:

• sorry I missed it in the initial submission, but can you explain the fatalities/death demonstrated in figure 1, and what is the purpuse of showing death, while you do not mention death otherwise in manuscript.

- We thank the reviewer for his feedback on this. In our initial manuscript, we did not include the deaths in the epidemiological curve as deaths were reported outside our enhanced surveillance period. However, as suggested by the reviewer as added the deaths in the graph. The authors agreed to remove the deaths as suggested by the academic reviewer as it is difficult to interpret (lines 100 to 103):

Fig 1. Epidemiology curve of reported AJS cases against number of facilities reporting (monthly average) in EWAR system; and timing of changes to the surveillance system from daily to weekly

• Another good point by one reviewed, also missed in first go-around, the title may best include jaundice as starting point, and that HAV was found as dominant culprit.

- We thank reviewer for the valuable suggestion. We have changed the title of our manuscript as suggested (lines 1 to 2) – 

“An outbreak of acute jaundice syndrome (AJS) among the Rohingya refugees in Cox’s Bazar, Bangladesh: findings from enhanced epidemiological surveillance.”

• Also, did you had an HCV IgM test available? I am not aware that such test exist.

- We thank the reviewer for pointing on the issue regarding HCV testing. During the epidemiological investigation, we used the “Biokit bioelisa HCV 4.0” ELISA kit, which uses anti HCV autoantibodies for detecting antigen against HCV from blood samples. The list of tests used with the name of the manufacturer is given below - 

No Disease/Pathogen Diagnostic test to be performed

1 Hepatitis A Virus HAV ELISA for IgM (CTK Biotech)

2 Hepatitis B Virus Hepatitis B surface antigen (RPC Diagnostics)

3 Hepatitis C Virus Anti-HCV antibodies (Biokit bioelisa HCV 4.0)

4 Hepatitis E Virus Anti-HEV ELISA for IgM (CTK Biotech & Euroimmun)

5 Leptospirosis Leptospira lateral flow for IgM (LifeAssay)

- We made the following changes in the manuscript (lines 168 to 169) – 

No Disease/Pathogen Diagnostic test to be performed

1 Hepatitis A Virus HAV ELISA for IgM 

2 Hepatitis B Virus Hepatitis B surface antigen 

3 Hepatitis C Virus Anti-HCV antibodies 

4 Hepatitis E Virus Anti-HEV ELISA for IgM

5 Leptospirosis Leptospira lateral flow for IgM

• I believe no one was HBV anti-HBc IgM positive?

- We only used Hepatitis B surface antigen (HBsAg) testing for the detection of HBV cases.

Minor compulsory revisions: 

All the comments on the paper need to be addressed as follows: 

A. Result section: 

1) In Table 3, Please mention which age group in first row??? 

 N (%) Odds Ratio (95% CI) P-value 

Age (in years) 275 (100%) 0.86 (0.83 to 0.89) < 0.001

This table shows two sub categories, age & gender. please make age data together and then put gender data 

- We thank reviewer for his valuable suggestion. We used age (in years) as numerical variable initially and age-group as categorical variable. However, we have made following changes in the manuscript to accommodate the suggested changes and merged 2 tables together to have same format (lines 256 to 259) – 

“Table 4. Association of hepatitis A seropositivity with age-group, gender, presenting symptoms and drinking water sources

 OR (95% CI) P-value

Age-group

0-4 years (n=67) 1.0* -

5-9 years (n=59) 1.15 (0.40 to 3.13) 0.79

10-17 years (n=60) 0.18 (0.07 to 0.42) < 0.001

18+ years (n=89) 0.02 (0.01 to 0.06) < 0.001

Gender

Female (n=119) 1.0* -

Male (n=156) 1.32 (0.82 to 2.14) 0.256

Presenting symptoms#

Fever (n=183) 2.25 (1.35 to 3.76) 0.002

Nausea (n=155) 1.19 (0.76 to 1.93) 0.479

Abdominal pain (n=113) 0.81 (0.50 to 1.31) 0.385

Vomiting (n=70) 1.85 (1.05 to 3.27) 0.034

Fatigue (n=57) 0.83 (0.46 to 1.50) 0.542

Itching (n=38) 1.23 (0.61 to 2.47) 0.563

Joint pain (n=37) 0.42 (0.21 to 0.86) 0.018

Loss of Appetite (n=27) 3.83 (1.40 to 10.45) 0.009

Dark Urine (n=26) 3.63 (1.32 to 9.94) 0.012

Bleeding (n=7) 1.04 (0.23 to 4.74) 0.960

Convulsion (n=6) 0.38 (0.07 to 2.12) 0.271

Drinking water sources^

Tube-well (n=170) 1.27 (0.78 to 2.06) 0.342

Tap water (n=45) 1.22 (0.63 to 2.33) 0.555

Unknown sources (n=58) 0.56 (0.32 to 1.01) 0.055

*Baseline category, #Diarrhoea (n=1) and other (n=2) symptoms were omitted from the summary table, ^ water truck (n=2) was omitted from the summary table”

In Table 4, pl put the data as per no. of cases presented with symptoms like fever most common presented (n=183) followed by Nausea (n=155) and abdominal pain (n=113) and so on.. 

- We thank reviewer for this suggestion on data presentation. We have made the changes in the manuscript as suggested lines (256 to 259). 

Map Section: 

1. AR should be calculated age-wise, sex wise and location wise.

- We thank reviewer for the valuable suggestion. We have calculated the attack rates and incorporated in the manuscript (lines 205 to 215) – 

“Camp-wise weekly attack rate (AR) per 10,000 population was calculated for the reported 275 AJS cases detected by the enhanced epidemiological surveillance strategy between 28 February and 26 March 2018. The weekly mean attack rate over the enhanced surveillance period (28 February to 26 March 2018) was 0.60 (95% CI: 0.28 to 0.92) per 10,000 population, across all the camps for the reported AJS cases. However, the AR varies among age-groups and was higher in younger age-groups (see Table 3). AR (per 10,000 population) for different age-groups is graphically presented in Appendix 5.

Table 3: Attack rate of reported AJS cases by age-group* and gender among all 34 camps between 28 February and 26 March 2018 in Cox’s Bazar, Bangladesh

 Weekly AR/10,000 population (95% CI)

Overall 0.60 (0.28 to 0.92)

By age-group

0-4 years 0.80 (0.34 to 1.26)

5-11 years 0.73 (0.36 to 1.10)

12-17 years 0.71 (0.31 to 1.11)

18+ years 0.43 (0.15 to 0.92)

By gender

Female 0.49 (0.21 to 0.77)

Male 0.71 (0.33 to 1.09)

* Population data from UNHCR Bangladesh Operational Update, 1 - 15 August 2018, was used to calculate the AJS attack rate (14).”

“

2. Map of camps are shown as no. of cases with different gradients, it should be depicted as per Attack rate (No of AJS cases in camp wise/ total population of given camp %), 

- We thank reviewer for the suggestion. We have created the following maps depicting attack rates in different camps (lines 100 to 103 and 454 to 457) – 

Fig 1. Epidemiology curve of reported AJS cases against number of facilities reporting (monthly average) in EWAR system; and timing of changes to the surveillance system from daily to weekly

Appendix 5. Map of AJS attack rate (per 10,000 population) by age-groups reported during enhanced epidemiological surveillance strategy (between 28 February and 26 March 2018) in the Rohingya refugee camps in Cox’s Bazar, Bangladesh “

3. We can see there were 34 camps and 18 health facilities, for better interpretation, map can be merged as per catering population by each health facility and calculate AR accordingly. 

- We thank the reviewer for his thoughtful suggestion. However, we do not have data on clinics’ catchment area as it is very hard to determine catchment area by facility in the camps. We used camp-wise population data to calculate attack rate per only. 

Discussion Section: 

1. As per topic heading is concerned, “Hepatitis A outbreak among the Rohingya refugees in Cox’s Bazar, Bangladesh:findings from enhanced epidemiological surveillance”; it’s better to keep “Jaundice outbreak among the Rohingya refugees in Cox’s Bazar, Bangladesh: findings from enhanced epidemiological surveillance”- there were cases multifactorial etiologies found after investigation, so in my opinion Jaundice outbreak covers all details. 

- We thank reviewer for his suggestion. We have amended our title as follows (lines 1 to 2) – 

“An outbreak of acute jaundice syndrome (AJS) among the Rohingya refugees in Cox’s Bazar, Bangladesh: findings from enhanced epidemiological surveillance.” 

2. Regarding Hep B cases found during surveillance, warrants immunization Hep- B drive in all age group healthy individuals in 34 camps as well as zero doses to new borne ASAP to protect from spread and close monitoring to cases. 

- We thank reviewer for this suggestion on the talking point for hepatitis B cases in the discussion section. We made the following changes as suggested by the reviewer (lines 321 to 331)– 

“While Hepatitis B and C were not drivers of the AJS epidemic, the detection of significant number of infections requires attention and likely reflects high prevalence of chronic infections. l. One small study has shown significant prevalence of HBV and HCV among the Rohingya population in Bangladesh (15) and warrants further serological investigation in particular given limited capacity for clinical management within the district. Hepatitis B vaccination is part of the routine EPI programmes in the camps, and our results highlight the need to ensure adequate coverage, as well as a zero dose to new-borns to protect from spread of the infection, with close monitoring of identified cases. There is also need to maintain routine immunization programmes, to improve vaccine coverage of Hepatitis B among women of reproductive age (WRA) and under-five children both in the camps and the surrounding host community (4). This might, in turn, reduce the perinatal transmission of hepatitis viruses (19).”

3. In last, I feel for better representation of data, descriptive data should be kept separate and analytical data should be separate. 

- We thank reviewer for this suggestion. We made necessary changes in the manuscript. 

Please send word documents more comments on track change mode. 

The paper could be improved by few editing and formatting. 

- We thank reviewer for this valuable suggestion. We also tried to improve the manuscript in our revised version. We have made some changes in the discussion segment to make it more focused given all the changes.

---

## [Editor Report · Decision Letter 2]

8 Apr 2021

An outbreak of acute jaundice syndrome (AJS) among the Rohingya refugees in Cox’s Bazar, Bangladesh:  findings from enhanced epidemiological surveillance

PONE-D-20-32697R2

Dear Dr. Mazhar,

We’re pleased to inform you that your manuscript has been judged scientifically suitable for publication and will be formally accepted for publication once it meets all outstanding technical requirements.

Kind regards,

Hans Tillmann

Academic Editor

PLOS ONE
---

## [Editor Report · Acceptance letter]

16 Apr 2021

PONE-D-20-32697R2 

An outbreak of acute jaundice syndrome (AJS) among the Rohingya refugees in Cox’s Bazar, Bangladesh:  findings from enhanced epidemiological surveillance. 

Dear Dr. Mazhar:

I'm pleased to inform you that your manuscript has been deemed suitable for publication in PLOS ONE. Congratulations! Your manuscript is now with our production department. 

Kind regards, 

on behalf of

Dr. Hans Tillmann 

Academic Editor

PLOS ONE